# Comparative analysis of extracellular vesicles from induced and adipose-derived Mesenchymal Stem Cells: Implications for regenerative medicine

Sura Nashwan[1,2], Mohammad A. Ismail[2,3], Tareq Saleh[4,5], Sabal Al Hadidi[2], Enas Alwohoush[2], Momen Sarhan[2,6], Nizar Abu Harfeil[1], Abdalla Awidi[2,6,7,8], Nidaa A. Ababneh[2]*

1 Department of Applied Biology, Jordan University of Science and Technology, Irbid, Jordan, 2 Cell Therapy Center, The University of Jordan, Amman, Jordan, 3 South Australian ImmunoGENomics Cancer Institute, Adelaide Medical School, University of Adelaide, Adelaide, South Australia, Australia, 4 Department of Pharmacology & Therapeutics, College of Medicine & Health Sciences, Arabian Gulf University, Manama, Bahrain, 5 Department of Pharmacology and Public Health, Faculty of Medicine, The Hashemite University, Zarqa, Jordan, 6 School of Medicine, The University of Jordan, Amman, Jordan, 7 Hemostasis and Thrombosis Laboratory, School of Medicine, The University of Jordan, Amman, Jordan, 8 Department of Hematology and Oncology, Jordan University Hospital, Amman, Jordan

* n.ababneh@ju.edu.jo, nidaaanwar@gmail.com

## Abstract

Extracellular vesicles (EVs), which include exosomes (Exos) and microvesicles (MVs), play a crucial role in intercellular communication and exert various biological activities by delivering specific cargoes of functional molecules, such as RNAs and proteins, to target cells. EVs secreted by human mesenchymal stem cells (hMSCs) have demonstrated their capacity to replace intact MSCs in tissue repair and regeneration. Induced mesenchymal stem cells (iMSCs) derived from induced pluripotent stem cells (iPSCs) present a promising alternative to traditional MSCs for producing EVs. This study aimed to establish an alternative source of EVs from iMSCs and compare them with EVs from adipose-derived MSCs (ADMSCs). Both iMSCs and ADMSCs were expanded under xeno-free culture conditions, and conditioned media were collected for EV isolation and characterization. The effects of the isolated EVs on cellular viability, apoptosis, senescence, and cell migration were evaluated. Results indicated that iMSC-EVs had a larger particle size (~1.5-fold) with no significant differences in morphology or surface markers compared to ADMSC-EVs. Furthermore, both iMSC- and ADMSC-derived EVs significantly increased HDF viability at 48 and 72 hours ($p \leq 0.01$, $p \leq 0.05$). Both types of EVs significantly reduced apoptosis levels ($p \leq 0.01$) in both HDFs and ADMSCs, while having no effect on senescence induction ($p > 0.9999$). Additionally, iMSC-EVs significantly enhanced ADMSC migration ($p < 0.0001$), whereas the effect was less pronounced with ADMSC-EVs. iMSC-EVs present a promising and a scalable option for regenerative applications,

**Data availability statement:** All relevant data are within the manuscript and its Supporting Information files.

**Funding:** This work was supported by the Deanship of Research at the Jordan University of Science and Technology [grant number 20210398] and the Deanship of Scientific Research at the University of Jordan (141/2020).

**Competing interests:** The authors have declared that no competing interests exist.

offering advantages over ADMSC-EVs. However, further investigation is needed to fully understand their effects and underlying mechanisms.

## Introduction

Mesenchymal stem cells (MSCs) are multipotent cells that exhibit plastic adherence, possess the ability to self-renew, and differentiate into a variety of cell types, including chondrocytes, osteocytes, adipocytes, as well as cardiac and neuronal cells [1,2]. MSCs are found in nearly all connective tissues [3] and are commonly utilized in regenerative medicine due to their immunomodulating, anti-apoptotic, and anti-fibrotic activities [4,5]. Although MSCs can differentiate into various cell types to repair damaged tissues, MSC transplantation carries certain risks, including the potential for tumor formation, low recovery rates, and the possibility of graft rejection [6,7].

The therapeutic potential of stem cells can be attributed to their indirect interaction with recipient somatic cells through the release of cytokines, chemokines, anti-inflammatory factors, and growth factors, as well as small vesicles called extracellular vesicles (EVs) [8–10]. EVs are diverse populations of membranous structures derived from cell membranes of all cell types [11]. Based on their size, EVs can be broadly categorized into two main groups, small and large EVs. Moreover, small EVs can be further divided into: exosomes (Exos) with size ranging between 30 to 200 nm and microvesicles (MVs) with size ranging from approximately ~100 to 1,000 nm [12]. EVs play a crucial role in cell-cell communication between local and distant cells, facilitating the transfer of biological content, including proteins, lipids, and nucleic acids [13–16].

EVs derived from human MSCs (hMSCs) have demonstrated superiority in maintaining similar functions to MSCs and avoiding apparent adverse effects [17]. Moreover, MSC-derived EVs (MSC-EVs) can replace intact MSCs in tissue repair and regeneration [18,19]. Consequently, MSC-EVs represent a new avenue for treating various conditions, such as tissue injuries [18,20], radiation damage to bone marrow hematopoietic cells [21], fractures [22], and neurodegeneration [23]. However, several challenges have been identified against the clinical application of MSC-EVs, including inefficient production, inadequate reliability, and purification of clinical grade EVs [20,24].

Given the stable nature of induced pluripotent stem cells (iPSCs), it is hypothesized that iPSCs might be a suitable target for EVs-mass production [25]. iPSCs are a subset of cells that can be reprogrammed from any human tissue type through the forced expression of pluripotency transcription factors [26]. iPSCs can be induced *in vitro* to generate induced MSCs (iMSCs) through embryoid body formation [27]. iMSCs can surpass most of the MSC limitations because they are (*i*) capable of producing a large number of cells with high purity, (*ii*) have a high proliferation rate with a short doubling time, (*iii*) have extended expansion capacity [28], (*iv*) could be a universal source preventing the heterogeneity of MSCs isolated from different sources [29] and (*v*) meet most of the international Society for Cell and Gene Therapy (ISCT) criteria for MSCs [30]. Compared to adult MSCs, iMSCs exhibited superiority in cell

proliferation, immunomodulation, and synthesis of exosomes capable of regulating the microenvironment. Exosomes derived from iMSCs might overcome immune rejection following cellular transplantation, providing an alternative approach to mitigate the potential risks that follow cellular therapy [31].

This study aimed to explore an alternative source of EVs derived from induced mesenchymal stem cells (iMSC-EVs) and to compare their physical properties and biological functions with EVs from adipose-derived MSCs (ADMSC-EVs). Additionally, the uptake of iMSC-EVs and ADMSC-EVs was assessed by different cell types. We further investigated the impact of the isolated EVs on cellular viability, apoptosis, senescence, and migration *in vitro*.

## Materials and methods

### Ethical approval and donor consent

Three ADMSC samples used in this study were isolated and expanded as described in our previous publications [32,33]. This study adhered to ethical guidelines and was approved by the Institutional Review Board of the University of Jordan and the Cell Therapy Center in 2019 (IRB-7-2019-7) and 2021 (IRB-CTC/2–2021/06). All three female donors provided signed written informed consent before their participation in the study. Adipose tissue samples were collected from healthy female patients with age range between 30–45 and no history of diseases. Donors were undergoing liposuction at the Plastic Surgery Department of Jordan University Hospital (JUH)/the University of Jordan between January 1, 2020, and January 1, 2022. Induced pluripotent stem cell (iPSC) lines were generated and characterized from dermal skin fibro-blasts, as detailed in our previous publication [34]. The pluripotency of these cell lines was confirmed via flow cytometry analysis for the expression of Nanog and Tra-1–60, following our previously established protocols [34].

### Generation of induced mesenchymal stem cells (iMSCs) via embryoid bodies (EBs)

To generate induced mesenchymal stem cells (iMSCs), we first generated embryoid bodies (EBs) from induced pluripotent stem cells (iPSCs) as described previously [32]. Three different iPSC lines were cultured on Matrigel (Corning)-coated plates and maintained in mTeSR (Stem Cell Technologies, Vancouver, Canada). Cells were detached using 1xTryplE (Gibco, New York, USA), and the cell suspensions were seeded in ultra-low attachment plates at a seeding density of $2x10^5$ cells/well. Cells were maintained in mTeSR supplemented with 10 μM Rock inhibitor (R&D, London, UK) for 24 hours to facilitate EB formation. On the second day, mTeSR was replaced with MSC complete culture media (CCM), composed of Minimum Essential Medium Eagle-Alpha Modification (aMEM) (Gibco, New York, USA) supplemented with 15% fetal bovine serum (FBS, Hyclone), 1% 100X Glutamax (Gibco, New York, USA), and 1% 100X antibiotic-antimycotic mixture (Gibco, New York, USA). On days 2 and 4 of differentiation, the media was replaced with fresh media supplemented with 10 μM retinoic acid (RA) and 0.1 μM RA, respectively (Sigma-Aldrich, Darmstadt, Germany) to promote differentiation. On day 6, the media was switched to an RA-free differentiation medium. On day 7, EBs were plated on Matrigel-coated plates and maintained in MSC differentiation media. The medium was exchanged every two days, and on day 12, the culture media was supplemented with 2.5 ng/ml basic fibroblast growth factor (bFGF) to enhance iMSCs proliferation and differentiation. Once iMSCs reached 80–90% confluency, they were passaged and cryopreserved in 1X freezing media (90% FBS and 10% DMSO) before storage in liquid nitrogen (LN). For comparison, original ADMSCs lines of four age-matched female donors, previously generated at the Cell Therapy Center (CTC), University of Jordan, Jordan, were used and expanded using the same CCM and culture conditions. All cell lines were expanded and maintained under standard culture conditions (37°C, 21% $O_2$, and 5% $CO_2$).

### Flow cytometry of iMSCs and ADMSC surface markers

To characterize iMSCs and ADMSCs, cells were assessed for the presence of hMSCs surface markers (CD90, CD105, CD73, and CD44) and the absence of hums' negative markers (CD34, CD45, CD11b or CD14, CD19 or CD79α, and HLA-DR) using the BD Stem Flow hMSC Analysis kit (BD Biosciences, California, USA). A cocktail of fluorescently

conjugated antibodies targeting hMSC-positive markers (CD90, CD105, CD73, and CD44) and their isotype controls were freshly prepared following the manufacturer's instructions. Samples were analyzed using the BD FACS Canto II flow cytometer and BD FACSDiva software.

### iMSCs and ADMSCs osteogenic and adipogenic differentiation

For osteogenic differentiation, cells were harvested and seeded in triplicates in 6-well tissue culture plates at $2 \times 10^5$ cells/well. Cells were maintained in CCM until reaching at least 50% confluency, and then media was replaced with a complete osteogenic differentiation medium composed of Minimum Essential Medium Eagle-Alpha Modification (Alpha MEM, Gibco) supplemented with 10% fetal bovine serum (FBS, Hyclone), 1% 100X Glutamax (Gibco, USA), 1% 100X antibiotic- antimycotic mixture (Gibco, USA), 10 mM dexamethasone, 50 µg/ml ascorbic acid 2-phosphate, and 10 mM β-glycerophosphate (Carbosynth, USA). Cells were maintained for 21–28 days until calcium deposition was observed. Control cells were kept in CCM for each sample, and media was exchanged every 2–3 days. Once mineral deposition was observed under the microscope, one well of each sample was used for Alizarin Red staining of calcium crystals, examined, and imaged using the EVOS XL Core Imaging System (Thermo Fisher, Waltham, Massachusetts) [32].

The same procedure described above was applied for adipogenic differentiation composed of Minimum Essential Medium Eagle-Alpha Modification (Alpha MEM, Gibco) supplemented with 10% fetal bovine serum (FBS, Hyclone), 1% 100X Glutamax (Gibco, USA), 1% 100X antibiotic- antimycotic mixture (Gibco, USA), 10 mM dexamethasone, 500 µM 3-isobutyl-1-methylxanthine (IBMX), 0.2 mM indomethacin, and 10 µg/ml insulin and maintained for 14–21 days. Once fat vacuoles were visible under the microscope, samples were stained with Oil Red-O stain and imaged using the EVOS XL Core Imaging System [32].

### EVs collection and characterization

Cells of passage 3 of both iMSCs and ADMSCs were cultured in a single-layer chamber (Corning, New York, USA) and maintained in CCM until they reached 80–90% confluence. Then, the medium was exchanged, cells were incubated in a serum-free medium (SFM) for 48 hours, and the conditioned medium (CM) was collected. Specifically, the CM underwent sequential centrifugation at 300 g for 10 minutes at 4 °C to eliminate the remaining cells, followed by centrifugation at 2000 g for 20 minutes at 4 °C to remove dead cells and large apoptotic bodies. Subsequently, cellular debris was eliminated through 0.22-µm filtration, and supernatants were subjected to ultracentrifugation at 110,000 g for 2 hours at 4 °C using the CS-FNX Micro Ultracentrifuge (Ibaraki, Japan). Then, the resulting EV pellets were resuspended in 500 µL of filtered 1X PBS, and their concentration was measured using the Micro BCA™ Protein Assay Kit (Thermo Scientific, USA) according to the manufacturer's guidelines. The isolated EVs were aliquoted, adjusted to a final concentration of 1 mg/mL, and stored at −80°C for future use.

Proteins of the tetraspanin family (CD9, CD81, and CD63) were studied to demonstrate the lipid-bilayer structure of EVs via beads-based flow cytometry. For each EV sample, three tubes were prepared: CD9 and CD81 together, CD63, and Unstained EVs. In each tube, 100 µl of each EV sample were incubated with 3 µl of Aldehyde/sulfate latex beads (4% w/v, 4 µm; Invitrogen, Massachusetts, USA) for 15 minutes at RT. Then 10 µl filtered PBS was added, and the EV-beads mixture was incubated overnight at 4°C under gentle movement (270° shaking). EV-beads binding was blocked by adding 10 µl of 1M glycine/tube and incubated for 30 minutes at RT. Subsequently, two samples were aliquoted in two test tubes; each tube contained of 50 µl of the EV-beads mixture and incubated for 40 minutes at 37°C. After that, 150 µl filtered PBS was added to each tube, and samples were processed on BD FACS Canto II, read at a low flow rate, and analyzed by BD FACSDiva software.

To examine the EV morphology, 100 µL (1 mg/ml) of purified EVs were resuspended in a 1:1 mixture with 2% PFA (paraformaldehyde) deposited in Formvar-carbon-coated electron microscope grids for 20 minutes in a dry environment at room temperature (RT). The grids were subsequently washed and immersed directly in drops of 1% glutaraldehyde and

incubated for 5 minutes. Then, seven washes with distilled water for 2 minutes each. To enhance contrast, the grids were placed on drops of uranyl-oxalate for 5 minutes, then transferred to methyl cellulose-UA for 10 minutes on ice. Finally, grids were air-dried for 10 minutes and examined using the Versa 3D FEI transmission electron microscopy (TEM) at an acceleration voltage of 30 V.

Size distribution was measured using a dynamic light scattering (DLS) nanosizer instrument (ZetaView, Malvern Nano ZS, Worcestershire, UK) to determine the particle sizes. Briefly, 100 µL (1 mg/ml) of EVs were resuspended in 500 µL of filtered PBS and properly mixed. Then, the data were analyzed using Zetasizer software 7.11, and the temperature was controlled at 24 °C. Data acquisition parameters were set as follows: Measurement angle (173° backscatter), number of runs (10), run duration (60 seconds), number of measurements (3), and the delay between measurements (10 seconds), based on the manufacturer`s recommendations for EV analysis.

### EVs labeling and cellular uptake

Cellular uptake of EVs was assessed by DiI fluorescent dye labelling (1,1′-dioctadecyl-3,3,3′,3′-tetramethylindocarbocyanine perchlorate; Invitrogen, Massachusetts, USA). In brief, approximately 200 µL (100 µg/mL) of iMSC-EVs or ADMSC-EVs were labeled with 5 µL of 10 µM DiI, a lipophilic dye that selectively stains EV membranes by integrating into their lipid bilayer. 1X PBS was used as a negative control and underwent the same incubation procedure with DiI. EVs were incubated with DiI dye in the dark for 1 hour with gentle mixing. Then, labelled EVs were resuspended in 4 ml of filtered PBS and subjected to ultracentrifugation at 110,000 g for 1 hour and 30 minutes at 4°C. Pellets were subsequently resuspended in 700 µl of filtered PBS.

Human fibroblasts cells were seeded on coverslips at a density of $1.5 \times 10^5$ cells/well and incubated with 30 µL of labeled EVs (100 µg/mL) from either iMSC-EVs or ADMSC-EVs for 12 hours at 37°C and 5% $CO_2$. Afterwards, cells were washed with PBS to remove unbound EVs and 5µl of 10µm CMFDA (5-chloromethylfluorescein diacetate, Invitrogen, Massachusetts, USA) in SFM was added and incubated for 30 minutes at 37°C to stain the cell body. Then, cells were fixed with 4% PFA for 15 minutes at RT. After that, cells were washed once with PBS and stained with DAPI (4′,6-diamidino-2-phenylindole, Invitrogen, Massachusetts, USA). Finally, cells were mounted with an anti-fade mounting medium (Abcam, Cambridge, UK), and images were captured under the lab observer microscope (Carl Zeiss, Oberkochen, Germany).

### Cell viability and apoptosis assays

To assess the variance in metabolic activity of HDF, cells were co-cultured with iMSC-EVs, MSC-EVs and HDF-derived EVs (HDF-EVs), a colourimetric (3-(4,5-Dimethyl-2-thiazolyl)- 2,5-diphenyl-2H-tetrazolium bromide (MTT), ATCC, Virginia, USA) was conducted. HDF cells were seeded into three 96-well plates at a seeding density of $8 \times 10^3$ cells/well in 100 µl CCM (media contains serum). After 24 hours, the cells were treated with 50 µg/ml of iMSC or ADMSC-EVs or HDF-EVs and SFM was used as a negative control. The following day, 10 µl of the MTT reagent was added to each well, and cells were incubated for 3 hours at 37°C. Subsequently, an equal volume of the solubilization stop solution was added to the wells (110 µl/well) and incubated at 37°C for 40 minutes. The absorbance was then recorded at 570 nm on Biotek Cytation 5 and analyzed using Bioteck Gen 5 data analysis software (BioTek, Vermont, USA). The same procedure was repeated for the other two plates after 48 and 72 hours.

Flow cytometry detected cell apoptosis using eBioscience® Annexin V-FITC Apoptosis Detection Kit (Invitrogen, Massachusetts, USA) following the manufacturer's instructions. HDF cells were seeded in 6 well-plates at a seeding density of $2 \times 10^5$ cells/well until they reached confluence. Then, the media was replaced with fresh SFM supplied with either 50 µg/ml iMSC, ADMSC-EVs, or SFM alone. After 48 hours, cells were harvested and washed with 1X PBS. They were resuspended in 100 µl 1X Binding Buffer and 5 µl FITC, followed by incubation in the dark for 15 minutes. After that, 5 µl of Propidium Iodide (PI) was added to each sample, along with 100 µl of 1X Binding Buffer to dilute the cell suspension. Samples were then directly assessed using BD FACS Canto II and analyzed by BD FACSDiva software.

## Senescence-associated β-Galactosidase analysis

Senescence-Associated β-Galactosidase (SA-β-Gal) staining was performed using a senescence detection kit (Abcam, cat #ab65351, Cambridge, UK). Briefly, HDFs were cultured on a 12-well plate, maintained in CCM at a seeding density of 1x10$^5$ cells/well and grown until they reached confluence. The complete media was aspirated, SFM was used, and the required quantity of EVs was added: 50 µg/ml of iMSC or ADMSC-EVs, and SFM served as a negative control. The cells were incubated at 37°C for 48 hours. After that, media were aspirated, and cells were washed once with 1 ml of 1X PBS and fixed with 0.5 ml fixative solution for 15 minutes at RT. Subsequently, 0.5 ml of staining solution mixture was added to each well and incubated at 37°C for 12 hours. Cells were observed under the microscope for the development of blue color.

## Scratch wound assay

To study the migration of HDFs and ADMSCs, a scratch wound assay was conducted on cells seeded on 6-well plates and grown until they reached confluence, then treated with 50 µg/ml of iMSC-EVs or ADMSCs-EVs, and SFM as a negative control for 24 hours. The cells were starved to distinguish between survival and migration, and a lesion was performed on the monolayer using a sterile 200 µm pipette tip. Then, cells were washed, and media were replaced with SFM-containing EVs, as described previously. Control wells with SFM only were also scratched and maintained in the same volume of SFM. After 0, 24, and 48 hours of adding EVs, the cells were imaged, and the cell migration distance was measured using ImageJ software.

## Statistical analysis

All data were analyzed on GraphPad Prism version 9.3.1 (GraphPad Software, California, USA) using unpaired students' t-tests or two-way analyses of variance (ANOVA) followed by the Bonferroni test when indicated. P-value $\leq 0.05$ was considered significant (* $\leq 0.05$, ** $\leq 0.01$, *** $\leq 0.001$, **** $\leq 0.0001$). All experiments were repeated at least in triplicate (n = 3).

## Results

### Characterization of iPSCs, iMSCs and ADMSCs surface markers and differentiation potential

Flow cytometry analysis confirmed the positive expression of pluripotency markers Nanog and Tra-1–60 in the iPSC lines used in this study (Supplementary S1 Fig). Further flow cytometry analysis demonstrated the successful differentiation of iPSCs into iMSCs, as indicated by the expression of hMSC surface markers (CD90, CD105, CD73, and CD44) in all iMSC samples, which was comparable to the ADMSCs samples. Additionally, both iMSC and ADMSC samples were negative for the MSC-negative markers cocktail (CD34, CD45, CD14, CD11b, CD79a, CD19, and HLA-DR) (Fig 1A & 1B). The differentiation potential of iMSCs and ADMSCs, a key feature for identifying and characterizing MSC populations, was further confirmed. Alizarin Red staining revealed that both iMSC and ADMSC samples successfully differentiated into the osteogenic lineage, as evidenced by calcium deposition (Fig 1C). However, Oil Red O staining showed a limited ability of iMSCs to differentiate into adipocytes, as indicated by the minimal fat vacuole formation (Fig 1D).

### Characterization of isolated EVs from iMSCs and ADMSCs

Flow cytometry analysis confirmed the successful expression of EVs surface markers in all analyzed samples. CD9 and CD81 were highly expressed, while CD63 exhibited relatively lower expression across three independent isolations of both iMSC-EVs and ADMSC-EVs. The average expression levels of EV surface markers in iMSC-EVs were as follows: CD9 (99.3%), CD81 (97.3%), and CD63 (77.3%). Similarly, ADMSC-EVs showed expression levels of CD9 (99%), CD81 (94%), and CD63 (65.3%) (Fig 2A and 2B). Overall, iMSC-EVs had slightly higher surface marker expression compared to ADMSC-EVs, though the difference was not statistically significant.

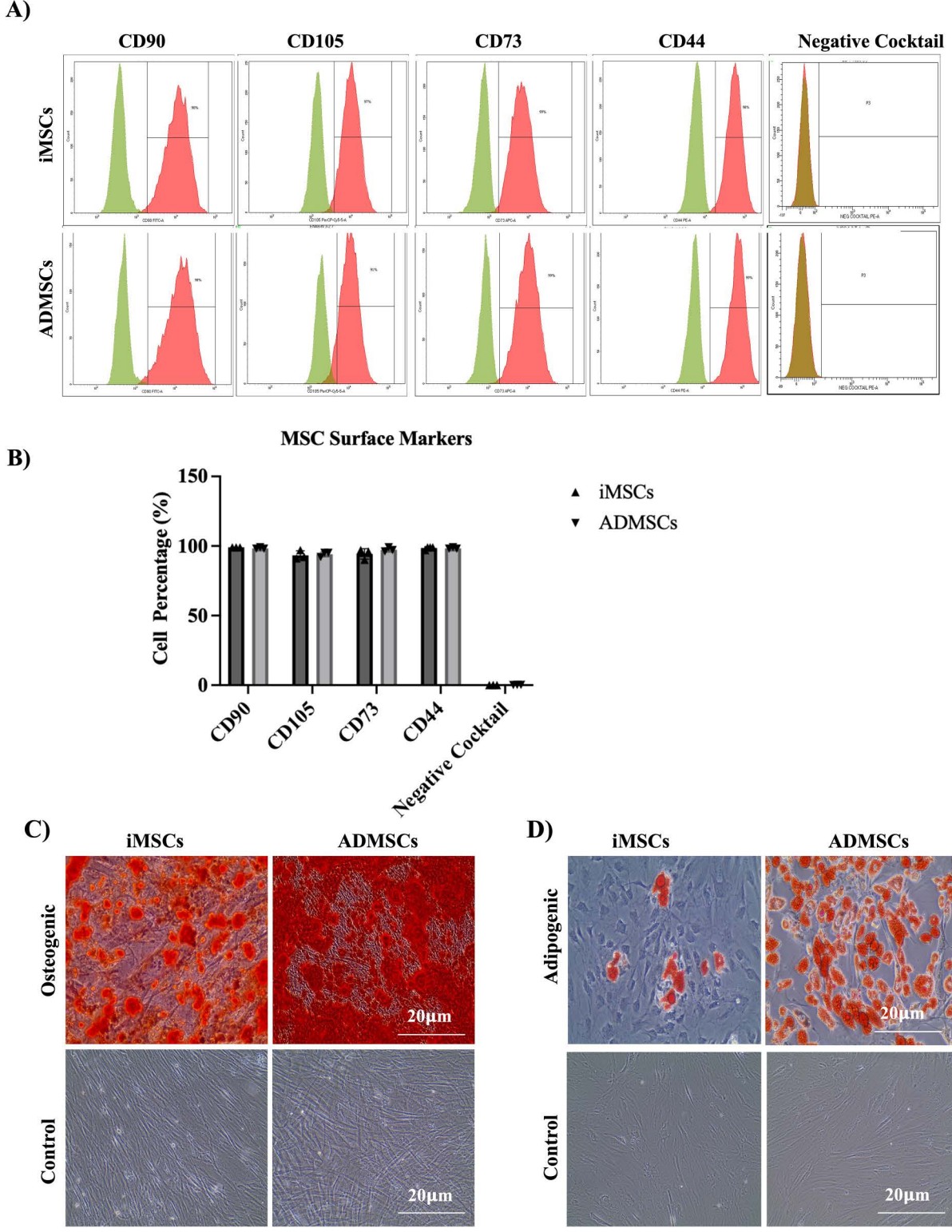

**Fig 1. Characterization of iMSCs and ADMSCs by flow cytometry and differentiation potential.** (A) Histograms of flow cytometric analysis showing the expression of positive human MSC surface markers and the negative markers cocktail for iMSCs and ADMSCs. (B) The percentages of iMSCs expressing the human MSCs markers compared to ADMSCs. (C) Alizarin Red staining of calcium deposits in the differentiated osteogenic cells. (D) Oil Red O staining of fat vacuoles in the adipocyte differentiated cells. Scale bar: 20 μm.

Additionally, DLS, a well-established, non-invasive technique for measuring particle size and size distribution in the submicron range, was used to analyze the size distribution of EVs. The analysis revealed that both iMSC-EVs and ADMSC-EVs exhibited heterogeneous size distributions, with two predominant particle populations. iMSC-EVs displayed size peaks at approximately 65 nm and 200 nm, while ADMSC-EVs showed peaks at around 30 nm and 172 nm. Additionally, iMSC-EV samples displayed readings above 5000 nm, indicating the presence of EV aggregates. Overall, iMSC-EVs were ~1.5 times larger than ADMSC-EVs (Fig 2C).

Finally, TEM revealed the typical cup-like morphology characteristic of both iMSC-EVs and ADMSC-EVs, consistent with their expected structure. TEM also confirmed the diameter measurements obtained from DLS and observed aggregations in some iMSC-EV samples (Fig 2D). These findings confirm the successful isolation of both iMSC-EVs and their parental ADMSC-EVs under the same culture conditions. Despite sharing similar characteristics, iMSC-EVs demonstrated a larger particle size compared to ADMSC-EVs.

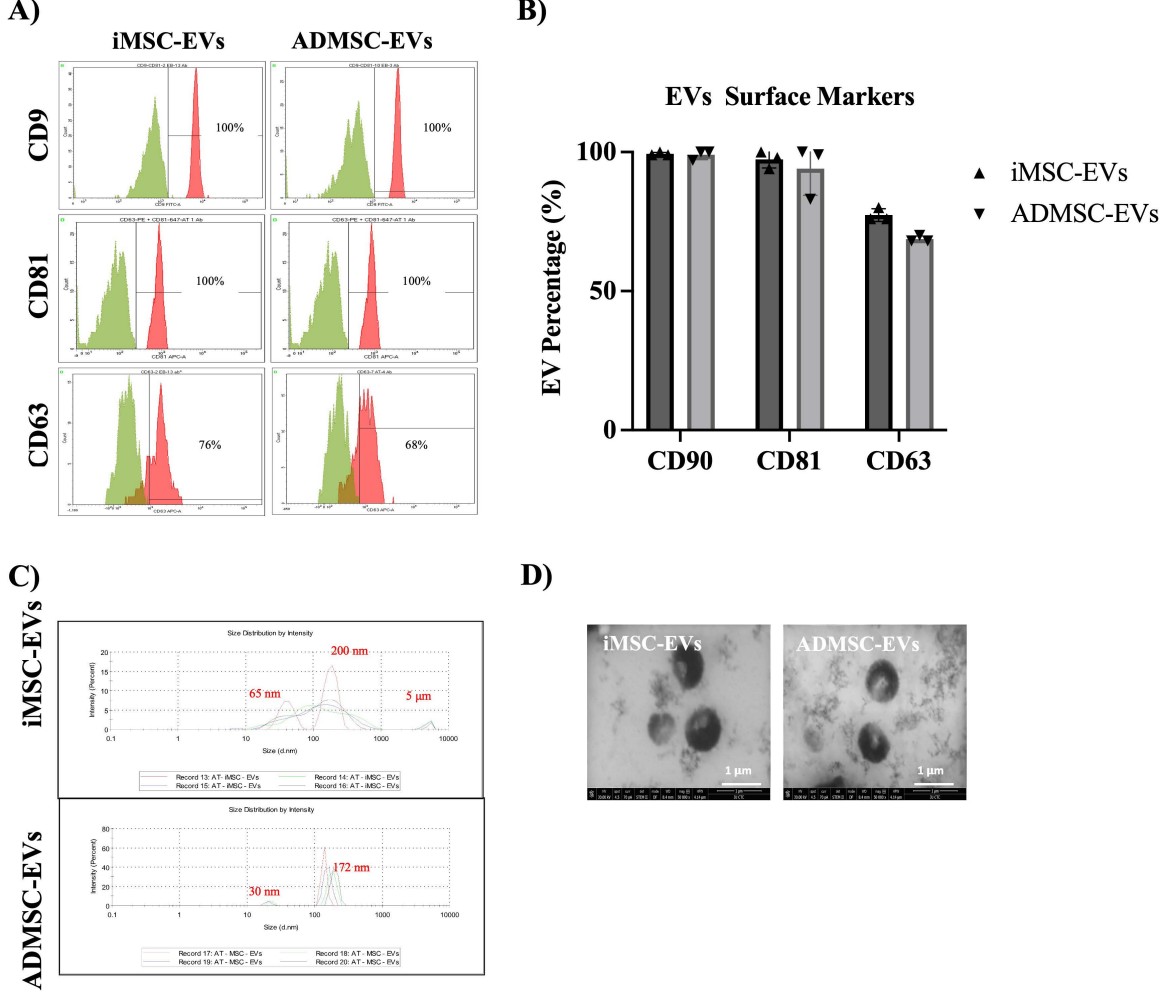

**Fig 2. Characterization of iMSC-EVs and ADMSC-EVs.** (A&B) Flow cytometry histograms and graph analysis of tetraspanin markers CD9, CD81, and CD63 on iMSC-EV and ADMSC-EV. (C) Size distribution of iMSC-EVs and ADMSC-EVs measured by DLS. (D) Representative images of iMSC-EVs and ADMSC-EVs by transmission electron microscopy. Scale bar: 1 μm.

## EVs were successfully internalized into HDF cells

To assess whether HDFs can internalize EVs, EVs were incubated with DiI, a fluorescent dye and lipophilic stain that integrates into lipid bilayer membranes and emits orange-red fluorescence. After 12 hours of incubation, cells were stained with CMFDA (green color), a fluorescent dye to monitor cell movement or location. Immunofluorescent images revealed red fluorescent particles in perinuclear regions, indicating that the EVs were taken up by HDF cells (Fig 3).

## iMSC-EVs enhanced HDF viability at 48 and 72 hours, while ADMSC-EVs show effects only at 72 hours

Following the confirmation of HDF cells' ability to internalize EVs, an MTT assay was conducted at 48- and 72-hours post-seeding to assess the impact of iMSC-EVs and ADMSC-EVs on cell viability. Notably, exposure to 50 µg/ml of iMSC-EVs significantly increased the viability of HDF cells compared to those treated with 50 µg/ml of ADMSC-EVs and HDF-EVs after 48 hours (p-value ≤ 0.01 and p-value ≤ 0.001, respectively) (Fig 4A). Additionally, after 72 hours, HDF cells treated with iMSC-EVs and ADMSC-EVs exhibited higher viability compared to the control SFM group (p-value ≤ 0.01 and p-value ≤ 0.05, respectively) (Fig 4A).

   Next, the Annexin V/propidium Iodide (PI) assay was used to evaluate apoptosis levels in HDF cells and ADMSCs (as a control) treated with 50 µg/ml of EVs derived from either iMSCs or ADMSCs. Flow cytometry analysis of Annexin V/PI staining revealed that both HDF cells (Fig 4B) and ADMSCs (Fig 4C) exposed to either iMSC-EVs or ADMSC-EVs exhibited significantly less apoptosis compared to cells treated with (SFM) (p-value ≤ 0.01 with all cell types and treatments).

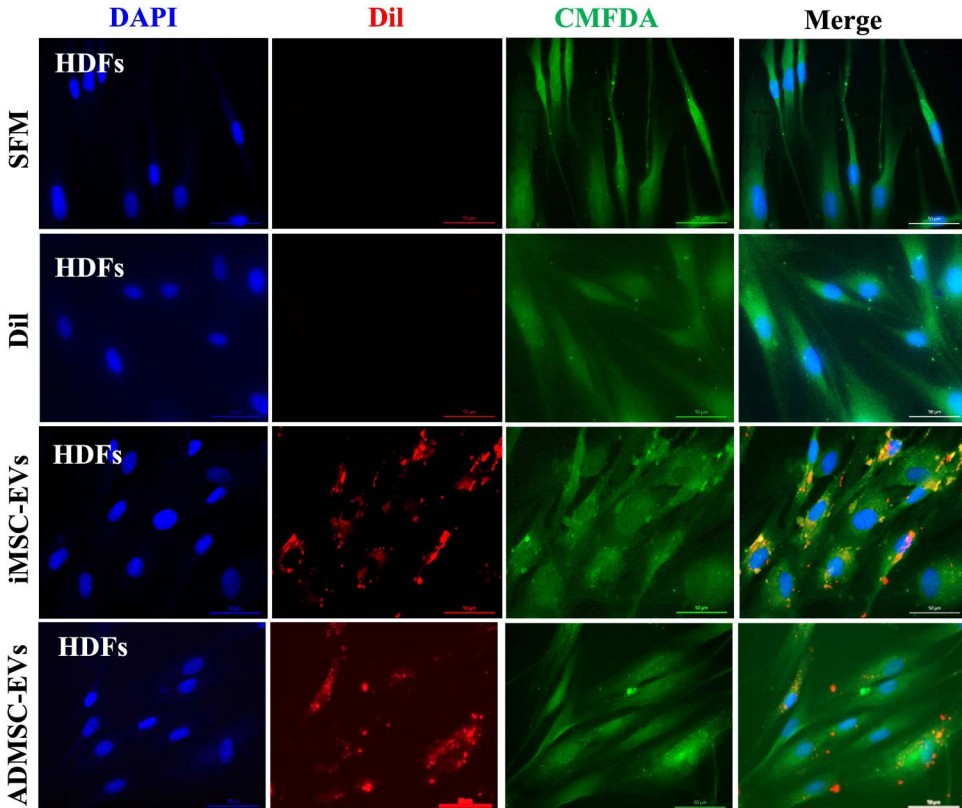

**Fig 3. iMSC-EVs and ADMSC-EVs are internalized by HDFs.** Representative images showing the uptake of DiI-labeled iMSC-EVs and ADMSC-EVs by human dermal fibroblasts (HDFs). DiI, a lipophilic dye, was used to stain EV membranes before incubation with HDFs. Scale bar: 50 µm.

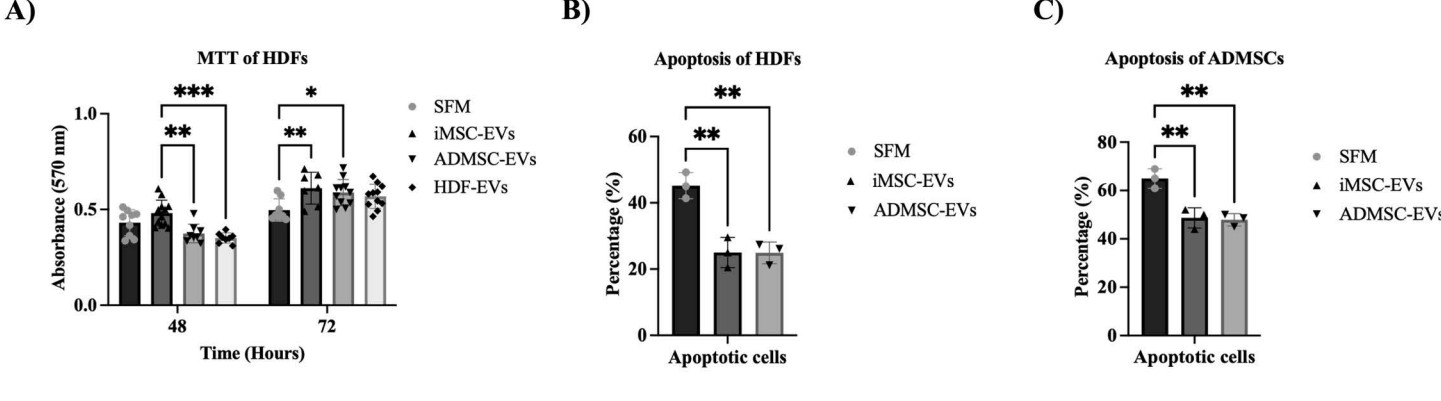

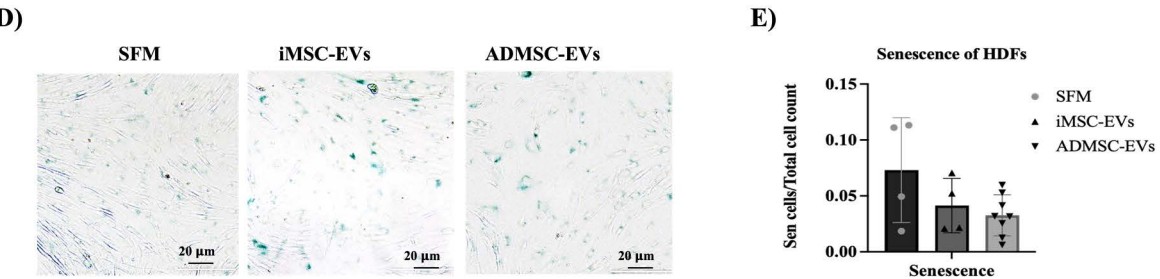

**Fig 4. Effect of iMSC-EVs and ADMSC-EVs on cell viability.** (A) MTT results of HDF samples treated with iMSC-EVs, ADMSC-EVs and HDF-EVs after 48 and 72 hours of cell seeding. (*p-value ≤ 0.05, ** p-value ≤ 0.01, ***p-value ≤ 0.001). The effect of iMSC-EVs and ADMSC-EVs on cell apoptosis of (B) HDF cells and (C) ADMSCs as illustrated by the graphs showing the percentage of apoptotic cells after treatment with iMSC-EVs and ADMSC-EVs (**p-value ≤ 0.01). (D) Representative images of SA β-Gal staining for senescent HDF cells in the presence or absence of iMSC-EVs and ADMSC-EVs. Scale bar: 200 µm. (E) The percentage of senescent cells after EVs treatment calculated as the number of SA-β-Gal positive cells divided by the numver of total cells.

These findings indicate that the increase in cell viability following EV exposure is accompanied by decreased cell death. Furthermore, cellular senescence, an additional stress response that halts proliferation, was examined. Senescence is characterized by a state of arrested growth, increased cell size, and upregulated SA-β-Gal activity, along with altered gene expression [35]. To determine whether EV exposure induced senescence in HDFs, SA-β-Gal staining was performed. However, no significant differences in SA-β-Gal staining were observed between the groups treated with 50 µg/ml of iMSC-EVs and ADMSC-EVs (p-value > 0.9999) (Fig 4D and 4E).

## EVs reduced ADMSC migration at 48 hours with minimal effect on HDFs

Cell migration plays a critical role in immune function and disease progression and is essential for tissue regeneration, as it enables the directed movement of cells to injury sites, promoting repair and tissue restoration [36]. The scratch wound assay revealed that treatment with 50 µg/ml of either iMSC-EVs and ADMSC-EVs significantly reduced the scratched area in ADMSC cultures compared to the SFM negative control after 48 hours (p-value < 0.0001). Moreover, iMSC-EVs demonstrated a significantly increased migration potential in ADMSCs than ADMSC-EVs after 48 hours (p-value < 0.0001) (Fig 5A and 5B). This suggests that EVs have an increased migration ability in ADMSC cultures. However, when HDF cells were treated with 50 µg/ml of iMSC-EVs or ADMSC-EVs, no significant differences in migration were observed compared to the negative control (SFM) (Fig 5C and 5D).

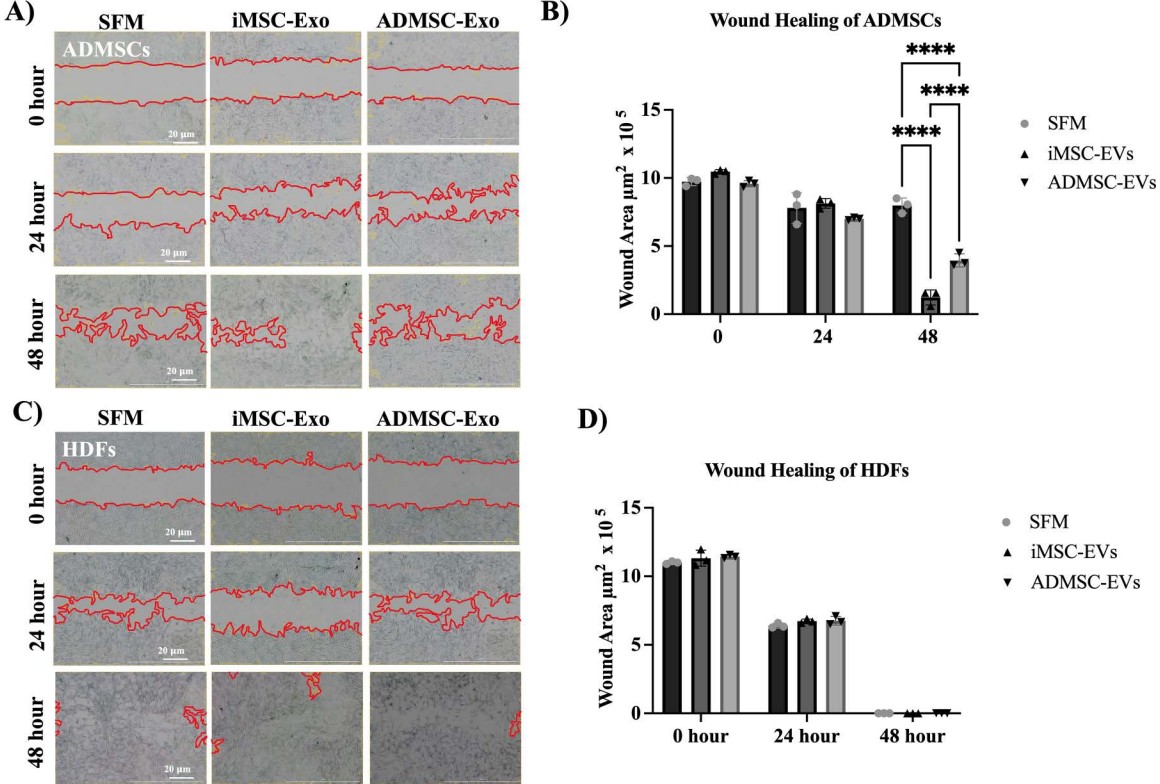

**Fig 5. The effect of iMSC-EVs and ADMSC-EVs on wound healing.** (A) Representative ADMSC images showing the scratched area at different time points 0, 24, and 48 hours (Evos, 4X magnification). Scale bar: 200 μm. (B) The quantification of scratched area μm2 in ADMSC cultures. (* p < 0.05, **** p-value <0.0001, ns-not significant). (C) Representative HDFs images showing the scratched area at 0, 24 and 48 hours (Evos, 4 X magnification) Scale bar: 200 μm. (D) The quantification of the scratched area in HDF cultures as a percentage.

## Discussion

The groundbreaking discovery of cellular reprogramming of somatic cells to the pluripotent state has paved the way for new approaches in regenerative medicine [37]. This innovation provides the advantages of producing iMSCs from iPSCs, emerging as a novel stem cell population that addresses many of the challenges associated with using MSCs for EVs and exosomes production [38,39]. Recent studies have highlighted the regenerative potential of exosomes derived from iMSCs, demonstrating their ability to promote angiogenesis, wound healing, and immune modulation [40,41]. [42].

In this study, iMSCs were generated from iPSCs, and EVs were isolated from iMSCs and compared to human ADMSC-EVs in terms of their characteristics and biological effects *in vitro*. The isolated iMSCs met the ISCT criteria for MSC specification [43]. Specifically, they exhibited plastic adherence, a fibroblast-like morphology, and expressed standard hMSC surface markers (CD90, CD105, CD73, and CD44). Additionally, both iMSC and ADMSC samples showed negative expression of MSC-negative markers, including CD34, CD45, CD14, CD11b, CD79a, CD19, and HLA-DR, confirming their MSC identity. Furthermore, they demonstrated the potential to differentiate into osteogenic and adipogenic lineages further validating their multipotency. While several studies have confirmed the ability of iMSCs to differentiate into osteoblasts, the adipogenic differentiation potential has been reported to range from poor to completely absent [44,45]. Tetraspanins, including CD81, CD63, and CD9, are widely recognized extracellular markers for EV characterization, as identified by the International Society for Extracellular Vesicles (ISEV) [46]. These markers were evaluated and confirmed to be positively expressed.

Additionally, we evaluated the size and morphology of both iMSC-EVs and ADMSC-EVs. DLS results indicated that iMSC-EVs had a larger particle size compared to ADMSC-EVs. EVs from both cell types fell within the typical size range of EVs/exosomes (30–200 nm). The heterogeneous sizes observed in both iMSC-EVs and ADMSC-EVs are expected, as they naturally fall within this range. Moreover, iMSC-EVs exhibited extracellular aggregates, which may have resulted from the high concentration of iMSC-EV samples or the ultracentrifugation process, a known limitation of this study. While this method yields a high concentration of small EVs, it can also contribute to aggregation [47]. Although this limitation does not affect the therapeutic relevance of our findings, it highlights the need for alternative or complementary approaches to improve EVs isolation. [48]. TEM analysis revealed that both EV groups displayed a cup-like shape with membranous structures, consistent with previous descriptions in the literature [46]. Together, these findings demonstrate that the isolation methods employed in this study for EVs are effective and allow for the analysis of nanoparticles in downstream applications.

The process of EVs cellular uptake is complex and remains an area of ongoing research [49,50]. Our findings indicate that both iMSC-EVs and ADMSC-EVs were effectively internalized by various cell models. In light of these results, there is potential to enhance the efficiency of EV cellular uptake. For example, Nakase et al. developed a method to improve cellular uptake of EVs by forming a complex with cationic lipids and a pH-sensitive fusogenic peptide. This approach may provide valuable insights for optimizing EV delivery in therapeutic applications [51]. Another study demonstrated cell-specific enhanced uptake by preloading albumin during the culture process to increase EVs uptake by liver cells [52].

We subsequently investigated the impact of EVs on the biological functions of the target cells in culture. Notably, the viability of cells treated with iMSC- EVs and ADMSC-EVs was significantly greater than the negative control, indicating their potential role in promoting cellular survival. At 72 hours, the increase in viability remained significant for both groups; however, iMSC-EVs exhibited a more pronounced effect, suggesting their prolonged pro-survival activity compared to ADMSC-EVs. These results align with previous findings demonstrating that human embryonic stem cell-derived EVs (hESC-EVs) can reduce infarct size in a mouse model of myocardial ischemia, further supporting the therapeutic potential of EV-based interventions in cardiac repair and regenerative medicine. The observed increase in cell viability may be attributed to EV-mediated activation of pro-survival pathways, such as the PI3K/Akt signaling cascade, which has been shown to enhance myocardial cell viability and inhibit adverse remodeling following cardiac injury [53,54]. These findings suggest that iMSC-EVs may provide comparable regenerative benefits by influencing key survival and repair mechanisms.

Interestingly, investigation of apoptosis revealed a significant reduction in the number of apoptotic cells following treatment with iMSC-EVs and ADMSC-EVs, in both HDF cells and ADMSCs. Both iMSC-EVs and ADMSC-EVs significantly reduced apoptosis compared to the SFM control, suggesting a prosurvival effect. When combined with the MTT assay results, this finding may indicate that the observed outcomes could be attributed to serum deprivation rather than the EV treatments. However, further studies are needed to investigate in more depth the mechanism behind these effects. Despite this limitation, our findings are in line with previous studies showing that MSC-derived EVs can inhibit apoptosis in neuronal cells, likely due to their cargo of anti-apoptotic proteins. Similarly, human umbilical cord MSC-EVs (hUCMSC-EVs) have been reported to provide hepatoprotection against acetaminophen (APAP)-induced toxicity, partly by inhibiting oxidative stress-mediated apoptosis via ERK1/2 and PI3K/Akt signaling [55,56]. These studies highlight the potential of stem cell-derived EVs as modulators of apoptosis and cell survival, although further mechanistic investigations are necessary to confirm their direct effects in our experimental model.

While we observed non-significant reductions in cellular senescence in HDF cells treated with both iMSC-EVs and ADMSC-EVs, literature suggests that MSC exosomes typically reduce senescence. For instance, Wang et al. showed that human fetal MSC secretome alleviates senescence in adult MSCs by reducing SA-β-Gal activity and promoting proliferation and osteogenic differentiation [57]. Similarly, MSC-derived supernatants modulate senescence in IL1β-treated osteoarthritis chondrocytes by regulating SA-β-Gal and reducing γH2AX foci and actin stress fibers [58]. However, our study did not reveal a significant reduction in the percentage of SA-β-Gal-positive cells. This discrepancy could be due

to differences in experimental conditions, cell types, or the limited range of senescence markers assessed. While both EV types showed a trend toward reducing senescence, these effects were less pronounced than those reported in other studies [59,60].

Wound healing is a dynamic and multifactorial process involving cellular proliferation, migration, and extracellular matrix remodeling [61]. Our results demonstrate that both iMSC-EVs and ADMSC-EVs significantly enhanced migration and wound closure in ADMSCs. However, no significant difference was observed in HDF migration across different conditions, suggesting that EV-mediated effects on wound healing may be cell-type specific. Surprisingly, this finding contrasts with a previous study indicating that treatment with human iPSC-EVs can facilitate cutaneous wound healing by promoting collagen synthesis and angiogenesis [43]. The observed discrepancies could be attributed to variability in MSC-derived EVs, influenced by factors such as the source of MSCs, donor variability, culture conditions, and EV isolation protocols [62–64]. These findings highlight the potential of iMSC-EVs as a promising tool for regenerative medicine, particularly in enhancing MSC migration and tissue repair. However, further research is required to elucidate the molecular mechanisms underlying their differential effects on various cell types and optimize their clinical applications [62–64].

The high yield of EV production from human iMSCs enhances their feasibility for laboratory research and potential clinical applications that require substantial amounts of purified EVs. Additional purification methods, such as anion exchange chromatography, could further validate our EV isolation protocol and enhance the overall purification process [24]. While iMSC-EVs have the advantage of being produced in larger quantities and promoting cell metabolism, EVs derived from other MSCs or cell types with specialized cargo may be more appropriate for certain therapeutic applications, such as stromal disorders in connective tissues [24].

However, this study did not evaluate the impact of isolated EVs on long-term cultures or measure the EV concentration in treated cells. Additionally, we did not quantify the total number of EV particles to compare yields between the two groups, which could be achieved using Nanoparticle Tracking Analysis (NTA). Further research is needed to validate our findings and explore the differences between EB-iMSC-EVs and other primary MSC-EVs. Given that EVs exhibit differential effects compared to previous studies, additional investigation is warranted to elucidate their mechanisms of action. Furthermore, assessing the effects of iMSC-EVs in disease models, such as cancer cell lines, is recommended.

## Conclusion

This study successfully isolated and characterized iMSC-EVs and ADMSC-EVs, revealing distinct extracellular properties. iMSC-EVs demonstrated higher uptake efficiency *in vitro* compared to ADMSC-EVs. Both types of EVs significantly enhanced cell viability and survival in HDFs and ADMSCs, with iMSC-EVs showing a stronger effect on ADMSC migration. These findings highlight the potential of iMSC-derived EVs as a scalable and reproducible alternative to MSC-derived EVs for regenerative applications. However, further research is needed to elucidate the molecular mechanisms driving these effects and to compare iMSC-EVs with other stem cell-derived EVs across different biological contexts.

## Supporting information

**S1 Fig. Pluripotency assessment of iPSC lines by flow cytometry.** The expression levels of TRA-1–60 and NANOG in three independent iPSC lines (iPSC1, iPSC2, iPSC3) were analyzed using flow cytometry. The green histograms represent the positively stained cells, while the gray histograms indicate the negative control. The percentage of positive cells is shown on each histogram.
(JPG)

**S2 Data. Raw data.**
(XLSX)

## Acknowledgments

We thank Prof Hatem Al-Kateib from the University of Jourdan/Faculty of Pharmacy for helping with DLS measurement and Rola Bqaien at the CTC/ University of Jordan for her assistance in TEM imaging of purified EVs.

## Author contributions

**Conceptualization:** Abdalla Awidi, Nidaa Ababneh.

**Formal analysis:** Sura Nashwan, Nidaa Ababneh.

**Funding acquisition:** Sura Nashwan.

**Methodology:** Sura Nashwan, Mohammad A. Ismail, Sabal Al Hadidi, Enas Alwohoush, Nidaa Ababneh.

**Project administration:** Nidaa Ababneh.

**Supervision:** Nizar Abu Harfeil, Abdalla Awidi, Nidaa Ababneh.

**Visualization:** Nizar Abu Harfeil.

**Writing – original draft:** Sura Nashwan.

**Writing – review & editing:** Tareq Saleh, Momen Sarhan, Nidaa Ababneh.

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
