## [Decision Letter · Decision Letter 0]

21 Jan 2025

PONE-D-24-47284Comparative Analysis of Extracellular Vesicles from Induced and Adipose-Derived Mesenchymal Stem Cells: Implications for Regenerative Medicine

PLOS ONE

Dear Dr. Ababneh,

Thank you for submitting your manuscript to PLOS ONE. After careful consideration, we feel that it has merit but does not fully meet PLOS ONE’s publication criteria as it currently stands. Therefore, we invite you to submit a revised version of the manuscript that addresses the points raised during the review process.

Please address and incorporate the following queries and suggestions, along with the concerns raised by the reviewer. "The section under this heading “Generation of Embryoid Bodies from iMSCs (EB-iMSCs)” should be divided into two distinct parts to improve clarity and flow. The second part should focus on the protocol for generating embryoid bodies (EBs) from induced Mesenchymal Stem Cells (iMSCs), including a detailed step-by-step process for embryoid body formation. This would include key factors such as cell seeding densities, culture conditions, and the duration of the induction period. It would be helpful to mention the media used  for example,  MSC differentiation medium used during the process, including any supplements or factors that are crucial for inducing pluripotency or differentiation into multiple germ layers. This would provide readers with a comprehensive understanding of the conditions required to generate functional MSCs.Additionally, to strengthen the claims of regenerative potential of exosome, further assays to assess the regenerative capabilities of exosomes derived from iMSCs should be included.

Regarding the figures, Figures 1C & D, Figure 2D, Figure 3D, and Figures 4A & C appear to be blurred and lack the resolution necessary for clear interpretation. These figures are crucial for supporting the results and should be reprocessed or replaced with higher-quality images to ensure clarity and improve the overall visual presentation. Enhanced clarity in these figures would make the data more convincing and allow for a better understanding of the experimental outcomes."

We look forward to receiving your revised manuscript.

Kind regards,

Dr Mahmood S Choudhery, PhD

Academic Editor

PLOS ONE

Journal Requirements:

“This work was supported by the Deanship of Research at the Jordan University of Science and Technology [grant number 20210398] and the Deanship of Scientific Research at the University of Jordan (141/2020).”

4. We notice that your supplementary figures are uploaded with the file type 'Figure'. Please amend the file type to 'Supporting Information'. Please ensure that each Supporting Information file has a legend listed in the manuscript after the references list.

**Additional Editor Comments:**

The section under this heading “Generation of Embryoid Bodies from iMSCs (EB-iMSCs)” should be divided into two distinct parts to improve clarity and flow. The second part should focus on the protocol for generating embryoid bodies (EBs) from induced Mesenchymal Stem Cells (iMSCs), including a detailed step-by-step process for embryoid body formation. This would include key factors such as cell seeding densities, culture conditions, and the duration of the induction period. It would be helpful to mention the media used for example, MSC differentiation medium used during the process, including any supplements or factors that are crucial for inducing pluripotency or differentiation into multiple germ layers. This would provide readers with a comprehensive understanding of the conditions required to generate functional MSCs.Additionally, to strengthen the claims of regenerative potential of exosome, further assays to assess the regenerative capabilities of exosomes derived from iMSCs should be included.

Regarding the figures, Figures 1C & D, Figure 2D, Figure 3D, and Figures 4A & C appear to be blurred and lack the resolution necessary for clear interpretation. These figures are crucial for supporting the results and should be reprocessed or replaced with higher-quality images to ensure clarity and improve the overall visual presentation. Enhanced clarity in these figures would make the data more convincing and allow for a better understanding of the experimental outcomes.

**Reviewers' comments:**

Reviewer's Responses to Questions

**Comments to the Author**

1. Is the manuscript technically sound, and do the data support the conclusions?

Reviewer #1: Yes

Reviewer #2: No

2. Has the statistical analysis been performed appropriately and rigorously? 

Reviewer #1: Yes

Reviewer #2: No

3. Have the authors made all data underlying the findings in their manuscript fully available?

Reviewer #1: Yes

Reviewer #2: Yes

4. Is the manuscript presented in an intelligible fashion and written in standard English?

Reviewer #1: Yes

Reviewer #2: Yes

5. Review Comments to the Author

Reviewer #1: Nashwan et al., have performed a comparative analysis of EVs isolated from IMSCs and ADMSCs. The authors need to work on the following comments to improve the manuscript.

1. Authors have demonstrated the role of EVs specifically in terms of mesenchymal cells and the generation of EVs from immune cells has not been discussed. Therefore, the following references are suggested to be cited in the third paragraph after reference #12

a. PMID: 36570199

b. PMID: 36713536

c. PMID: 38753658

2. Authors are suggested to incorporate these studies, which have indicated the negative role of MSC in bone cancer formation, along with reference 6.

a. PMID: 36769180

3. Authors are suggested to explain the findings of senescence data, if there is a significant difference between the SFM and ADMS-EVs in apoptosis and MTT data then why is it not a case of senescence?

4. The discussion section is again solely dedicated to mesenchymal cells, it should be broadly discussed in terms of EVs.

Reviewer #2: Ababneh and colleagues have explored the potential of extracellular vesicles (EVs) derived from induced mesenchymal stem cells (iMSCs) as an alternative to EVs from adipose-derived mesenchymal stem cells (ADMSCs) for regenerative medicine. The study compares iMSC-EVs and ADMSC-EVs in terms of size, effects on cell viability, apoptosis, migration, and senescence in vitro culture conditions. The authors argue that iMSC-EVs demonstrate a larger particle size and superior performance in enhancing cell viability and migration, while showing comparable effects on senescence and apoptosis reduction to ADMSC-EVs. Despite its innovative premise, the manuscript suffers from methodological inconsistencies, insufficient data to support key claims, unclear controls, and low-quality figure presentation. The manuscript thus requires major revisions to improve clarity, address limitations, and provide additional data where needed. These changes will strengthen the manuscript and enhance its overall impact.

Specific Comments:

1) EV regenerative properties: The main goal of the paper was to prove regenerative and wound healing properties of EVs derived from induced MSCs. However, it is not clear if these properties are general for any EVs. It would be thus necessary to include a comparison with control EVs (for example from Human Dermal Fibroblasts used in the experiments) that should lack wound-healing properties to establish the specificity of the observed effects.

2) Mis-cited references: The introduction lacks adequate references to support the claims. Mis-cited and irrelevant references make it difficult to verify key statements. Ensure that all references are accurate, recent, and contextually relevant. Here are a couple of examples for consideration:

a) Introduction, 3rd paragraph. The reference #13 does not support the statement that EVs are superior than MSCs. “EVs derived from human MSCs (hMSCs) have demonstrated superiority in maintaining similar functions to MSCs and avoiding apparent adverse effects [13].”

b) The reference #14 does not contain any mention of EVs. “Moreover, MSC-derived EVs (MSC-EVs) can replace intact MSCs in tissue repair and regeneration [14].”

c) Address inconsistencies in reference style (e.g., "(Van Niel, D’Angelo and Raposo, 2018)" vs. numbered citations).

3) EV dose used: The description of EV concentrations (10–50 µg/mL) provided in the methods section is not consistent with the results. Specify which concentration was used for each figure and experiment.

4) Figure quality and resolution: All figures need substantial improvement in quality. Microscopy images (Figures 1C, 1D, 5A, 5C) are unclear and need higher resolution. Axes, labels, and legends (e.g., Figure 4B) must be readable and clearly described.

5) EV size inconsistency: Discuss the inconsistency and heterogeneity in iMSC-EV size profiles observed in Figure 2C.

6) Data absent: Please provide data supporting the following statement for Figure 1. “Additionally, all iMSC samples were negative for the MSC- negative markers (CD34, CD45, CD14, CD11b, CD79a, CD19, and HLA-DR).”

7) Spontaneous apoptosis: Address the reason for high apoptosis levels in Figure 4C and clarify if the media conditions (e.g., absence of serum) caused this effect. Explore alternative controls like EVs derived from human dermal fibroblasts that do not protect from apoptosis.

8) Figure 3 description: Inconsistent description of experiment. Unclear if EVs or cells were stained with Dil.

a) In methods “iMSCEVs, MSCs-EVs, and PBS were labelled with DiI fluorescent dye selectively stained the plasma membrane and lipids.” Whereas in Results “To assess if HDFs can internalize EVs, cells were incubated with Dil as a fluorescent dye and a lipophilic stain, integrated into lipid bilayer membranes, and emitted orange-red fluorescence”.

b) The merged image of ADMSC-EVs does not show the red-Dil labeling staining.

c) The following statement lacks quantitative data for support. Please do quantitative image analysis to provide stronger proof. “The internalization efficiency of iMSC-EVs was higher than that of ADMSC-EVs.”

6. PLOS authors have the option to publish the peer review history of their article (what does this mean? ). If published, this will include your full peer review and any attached files.

**Do you want your identity to be public for this peer review?** For information about this choice, including consent withdrawal, please see our Privacy Policy .

Reviewer #1: **Yes: ** NAMRATA ANAND

Reviewer #2: No

---

## [Author Response · Author response to Decision Letter 1]

16 Mar 2025

PONE-D-24-47284

Comparative Analysis of Extracellular Vesicles from Induced and Adipose-Derived Mesenchymal Stem Cells: Implications for Regenerative Medicine

Please address and incorporate the following queries and suggestions, along with the concerns raised by the reviewer. "The section under this heading “Generation of Embryoid Bodies from iMSCs (EB-iMSCs)” should be divided into two distinct parts to improve clarity and flow. The second part should focus on the protocol for generating embryoid bodies (EBs) from induced Mesenchymal Stem Cells (iMSCs), including a detailed step-by-step process for embryoid body formation. This would include key factors such as cell seeding densities, culture conditions, and the duration of the induction period. It would be helpful to mention the media used for example, MSC differentiation medium used during the process, including any supplements or factors that are crucial for inducing pluripotency or differentiation into multiple germ layers. This would provide readers with a comprehensive understanding of the conditions required to generate functional MSCs. Additionally, to strengthen the claims of regenerative potential of exosome, further assays to assess the regenerative capabilities of exosomes derived from iMSCs should be included.

Response: Thank you for your valuable feedback. We have revised the section titled "Generation of Embryoid Bodies from iMSCs (EB-iMSCs)" by dividing it into two distinct parts for better clarity and flow. The first part is the Ethical Approval and Donor Consent, and the second part is the Generation of Induced Mesenchymal Stem Cells (iMSCs) via Embryoid Bodies (EBs). The first part provides a clear introduction to the protocol, while the second part includes a detailed, step-by-step process for embryoid body (EB) formation from induced mesenchymal stem cells (iMSCs). We have also incorporated the following details: Cell seeding densities, culture conditions, duration of induction period, media used (e.g., MSC differentiation medium), supplements or factors required for inducing pluripotency or differentiation into multiple germ layers.

We agree that further investigation of the regenerative capabilities of exosomes would largely strengthen the manuscript. However, this would have been tested in a supplemental project that is beyond the funding and logistical potential of the current manuscript. We are keen for the Editor’s understanding that conducting such state-of-the-art work, while essential, is not currently feasible. However, and to accommodate this concern, we expanded the Discussion section by incorporating recent studies demonstrating the regenerative potential of iMSC-derived exosomes in angiogenesis, tissue repair, and immune modulation. Additionally, we have addressed future directions by suggesting additional functional assays that could further validate the therapeutic potential of iMSC-derived exosomes. We hope that these modifications are somewhat satisfactory to this concern.

Regarding the figures, Figures 1C & D, Figure 2D, Figure 3D, and Figures 4A & C appear to be blurred and lack the resolution necessary for clear interpretation. These figures are crucial for supporting the results and should be reprocessed or replaced with higher-quality images to ensure clarity and improve the overall visual presentation. Enhanced clarity in these figures would make the data more convincing and allow for a better understanding of the experimental outcomes."

Response: Thank you for your valuable feedback. We appreciate the importance of ensuring that all figures are clear and of high resolution for proper interpretation. We have reprocessed Figures 1C & D, Figure 2 C & D , Figure 3 and Figures 4A & C to enhance their clarity and resolution. We hope the updated figures now meet the journal’s image quality requirements, and the data are presented in a more visually interpretable manner.

Response: The revised, marked-up and clean, versions of the manuscript have been submitted along with the point-by-point responses to all reviewers’ comments below.

Response to Editor’s comments

Response: Thank you for your guidance. We have reviewed and updated the manuscript to meet PLOS ONE's style requirements, including file naming.

“This work was supported by the Deanship of Research at the Jordan University of Science and Technology [grant number 20210398] and the Deanship of Scientific Research at the University of Jordan (141/2020).”

Response: Thank you for your guidance regarding the financial disclosure statement. We confirm that the funders had no role in the study’s design, data collection and analysis, decision to publish, or preparation of the manuscript. Accordingly, we have amended the financial disclosure to read: "The funders had no role in study design, data collection and analysis, decision to publish, or preparation of the manuscript."

Response: Thank you for your feedback regarding the Data Availability Statement. We acknowledge the journal’s requirements for sharing the minimal data set. In response to that, we did the following:

1. We revised Data Availability Statement in the manuscript to align with journal policies: “All relevant data are within the manuscript and its Supporting Information files..”

2. Graph Modifications: We have updated all graphical representations to display individual replicate values alongside summary statistics to enhance transparency. To ensure full reproducibility, we have also submitted the GraphPad Prism files containing raw data, statistical analysis details, and figure generation settings.

4. We notice that your supplementary figures are uploaded with the file type 'Figure'. Please amend the file type to 'Supporting Information'. Please ensure that each Supporting Information file has a legend listed in the manuscript after the references list.

Response: Thank you for pointing this out. We have resubmitted the supplementary figures under the ‘Supporting Information’ category and ensured that the file has a corresponding legend listed at the end of the manuscript in a section titled “Supporting information” after the references.

Reviewers' comments:

Review Comments to the Author

Reviewer #1:

Nashwan et al. have performed a comparative analysis of EVs isolated from iMSCs and ADMSCs. The authors need to work on the following comments to improve the manuscript.

1. Authors have demonstrated the role of EVs specifically in terms of mesenchymal cells and the generation of EVs from immune cells has not been discussed. Therefore, the following references are suggested to be cited in the third paragraph after reference #12

a. PMID: 36570199

b. PMID: 36713536

c. PMID: 38753658

Response: Thank you for your suggestions. We have incorporated the suggested references into the third paragraph of the Introduction section, following reference #12. The suggested references, now numbered as 14, 15, and 16, provide additional context on the generation of EVs from immune cells.

2. Authors are suggested to incorporate these studies, which have indicated the negative role of MSC in bone cancer formation, along with reference 6.

a. PMID: 36769180

Response: Thank you for the suggestion. We have incorporated the study (PMID: 36769180) into the manuscript, discussing the negative role of MSCs in bone cancer formation, and it is now numbered as reference #7, alongside with reference #6 as suggested.

3. Authors are suggested to explain the findings of senescence data, if there is a significant difference between the SFM and ADMS-EVs in apoptosis and MTT data then why is it not a case of senescence?

Response: Thank you for your important observation. We have updated the Discussion section on senescence to state the following:

“While we observed non-significant reductions in cellular senescence in HDF cells treated with both iMSC-EVs and ADMSC-EVs, literature suggests that MSC exosomes typically reduce senescence. For instance, Wang et al. showed that human fetal MSC secretome alleviates senescence in adult MSCs by reducing SA-βGal activity and promoting proliferation and osteogenic differentiation [57]. Similarly, MSC-derived supernatants modulate senescence in IL1β-treated osteoarthritis chondrocytes by regulating SA-βGal and reducing γH2AX foci and actin stress fibers [58] . However, our study did not observe significant reductions in the percentage of SA-βGal-positive cells. This discrepancy could be due to differences in experimental conditions, cell types, or the limited range of senescence markers assessed. While both EV types showed a trend toward reducing senescence, these effects were less pronounced than those reported in other studies [59], [60].”

4. The discussion section is again solely dedicated to mesenchymal cells, it should be broadly discussed in terms of EVs.

Response: Thank you for your valuable feedback. In response, we have revised the Discussion section to broaden the scope beyond mesenchymal stem cell-derived EVs (MSC-EVs) and provide a more comprehensive perspective on extracellular vesicles (EVs) from different cell types.

Reviewer #2:

Ababneh and colleagues have explored the potential of extracellular vesicles (EVs) derived from induced mesenchymal stem cells (iMSCs) as an alternative to EVs from adipose-derived mesenchymal stem cells (ADMSCs) for regenerative medicine. The study compares iMSC-EVs and ADMSC-EVs in terms of size, effects on cell viability, apoptosis, migration, and senescence in vitro culture conditions. The authors argue that iMSC-EVs demonstrate a larger particle size and superior performance in enhancing cell viability and migration, while showing comparable effects on senescence and apoptosis reduction to ADMSC-EVs. Despite its innovative premise, the manuscript suffers from methodological inconsistencies, insufficient data to support key claims, unclear controls, and low-quality figure presentation. The manuscript thus requires major revisions to improve clarity, address limitations, and provide additional data where needed. These changes will strengthen the manuscript and enhance its overall impact.

Response: We highly appreciate the time and effort by the reviewer to deliver this thorough and constructive feedback. We hope that the modifications conducted in the light of these comments have now greatly improved the quality of the work.

Specific Comments:

1) EV regenerative properties: The main goal of the paper was to prove regenerative and wound healing properties of EVs derived from induced MSCs. However, it is not clear if these properties are general for any EVs. It would be thus necessary to include a comparison with control EVs (for example from Human Dermal Fibroblasts used in the experiments) that should lack wound-healing properties to establish the specificity of the observed effects.

Response: The reviewer raises a very interesting point. We acknowledge the importance of establishing whether the regenerative and wound-healing properties observed in iMSC-derived EVs are specific or represent a general effect of EVs. While we did not initially include a direct comparison with control EVs, we have now incorporated additional data to the MTT from Human Dermal Fibroblast-derived EVs (HDF-EVs) to provide a broader perspective.

Due to experimental limitations, HDF-EVs were not tested in other functional assays (e.g., apoptosis and wound healing). The MTT assay was chosen as the primary comparison because it allows for a general assessment of EV-induced changes in cell viability, which serves as an indirect measure of regenerative potential. Future studies will be required to further explore the functional effects of HDF-EVs in additional assays, such as migration and apoptosis, to comprehensively assess their role. We are keen on the reviewer’s kind understanding on this issue.

2) Mis-cited references: The introduction lacks adequate references to support the claims. Mis-cited and irrelevant references make it difficult to verify key statements. Ensure that all references are accurate, recent, and contextually relevant. Here are a couple of examples for consideration:

a) Introduction, 3rd paragraph. The reference #13 does not support the statement that EVs are superior than MSCs. “EVs derived from human MSCs (hMSCs) have demonstrated superiority in maintaining similar functions to MSCs and avoiding apparent adverse effects [13].”

b) The reference #14 does not contain any mention of EVs. “Moreover, MSC-derived EVs (MSC-EVs) can replace intact MSCs in tissue repair and regeneration [14].”

c) Address inconsistencies in reference style (e.g., "(Van Niel, D’Angelo and Raposo, 2018)" vs. numbered citations).

Response: Thank you for your careful assessment of the citations. We have thoroughly reviewed the Introduction section and made the following revisions to ensure that all references are accurate, recent, and contextually relevant:

- Corrected Mis-Cited References.

- Ensured Citation Consistency.

- Strengthened References in the Introduction, by adding recent references in the introduction and discussion.

References adde

---

## [Decision Letter · Decision Letter 1]

7 May 2025

Comparative Analysis of Extracellular Vesicles from Induced and Adipose-Derived Mesenchymal Stem Cells: Implications for Regenerative Medicine

PONE-D-24-47284R1

Dear Dr. Ababneh,

We’re pleased to inform you that your manuscript has been judged scientifically suitable for publication and will be formally accepted for publication once it meets all outstanding technical requirements.

Kind regards,

Mahmood S Choudhery, PhD

Academic Editor

PLOS ONE

Additional Editor Comments (optional):

Reviewers' comments:

Reviewer's Responses to Questions

**Comments to the Author**

1. If the authors have adequately addressed your comments raised in a previous round of review and you feel that this manuscript is now acceptable for publication, you may indicate that here to bypass the “Comments to the Author” section, enter your conflict of interest statement in the “Confidential to Editor” section, and submit your "Accept" recommendation.

Reviewer #1: All comments have been addressed

Reviewer #3: All comments have been addressed

2. Is the manuscript technically sound, and do the data support the conclusions?

Reviewer #1: Yes

Reviewer #3: Yes

3. Has the statistical analysis been performed appropriately and rigorously? 

Reviewer #1: Yes

Reviewer #3: I Don't Know

4. Have the authors made all data underlying the findings in their manuscript fully available?

Reviewer #1: Yes

Reviewer #3: Yes

5. Is the manuscript presented in an intelligible fashion and written in standard English?

Reviewer #1: Yes

Reviewer #3: Yes

6. Review Comments to the Author

Reviewer #1: Authors are suggested to check the reference style all over the manuscript aftrer implementing the studies and make it according to the journal recomendations.

Reviewer #3: The authors have made the necessary corrections by adhering to the reviewers' opinions.

The authors addressed the Reviewers' requests carefully and corrected them.

The authors enriched the Manuscript.

The work is valuable and can significantly contribute to the scientific community in regenerative applications.

7. PLOS authors have the option to publish the peer review history of their article (what does this mean? ). If published, this will include your full peer review and any attached files.

**Do you want your identity to be public for this peer review?** For information about this choice, including consent withdrawal, please see our Privacy Policy .

Reviewer #1: **Yes: ** NAMRATA ANAND

Reviewer #3: No

---

## [Editor Report · Acceptance letter]

PONE-D-24-47284R1

PLOS ONE

Dear Dr. Ababneh,

I'm pleased to inform you that your manuscript has been deemed suitable for publication in PLOS ONE. Congratulations! Your manuscript is now being handed over to our production team.

Kind regards,

on behalf of

Dr. Mahmood S Choudhery

Academic Editor

PLOS ONE